# Association between Endometriosis, Irritable Bowel Syndrome and Eating Disorders: ENDONUT Pilot Study

**DOI:** 10.3390/jcm11195773

**Published:** 2022-09-29

**Authors:** Alexandra Aupetit, Sébastien Grigioni, Horace Roman, Moïse Coëffier, Amélie Bréant, Clotilde Hennetier, Najate Achamrah

**Affiliations:** 1Department of Gastroenterology, Rouen University Hospital, 37 Boulevard Gambetta, 76000 Rouen, France; 2Department of Nutrition, Rouen University Hospital, 37 Boulevard Gambetta, 76000 Rouen, France; 3INSERM UMR 1073 «Nutrition, Inflammation and Gut–Brain Axis Dysfunction», Normandie University, 76000 Rouen, France; 4Clinical Investigation Center CIC 1404, INSERM, Rouen University Hospital, 76000 Rouen, France; 5Multidisciplinary Franco-European Institute of Endometriosis, Clinique Tivoli-Ducos, 91 Rue de Rivière, 33000 Bordeaux, France; 6Department of Gynecology, Rouen University Hospital, 37 Boulevard Gambetta, 76000 Rouen, France

**Keywords:** irritable bowel syndrome, eating disorders, endometriosis

## Abstract

Background and aim: Irritable bowel syndrome (IBS), eating disorders (ED) and endometriosis share common pathophysiological mechanisms, involving alterations of the gut–brain axis. The aim of the ENDONUT pilot study was to investigate an association between these three diseases by screening for IBS and ED in patients with endometriosis. Method: We included patients from the CIRENDO cohort (Inter-Regional North-West Cohort of women with ENDOmetriosis) with a recent documented diagnosis of endometriosis of less than 4 years, regardless of age, date of onset of symptoms, type of endometriosis (digestive or not), with or without endometriosis-related digestive surgery. Validated questionnaires were used to screen for IBS (Rome IV, Francis score), ED (SCOFF-F, EAT-26), and anxiety/depression (HAD). Anthropometric data and lifestyle habits were also collected. The primary composite endpoint was SCOFF-F and ROME-IV scores. Results: Among 100 patients meeting inclusion criteria, 54 patients completed all the questionnaires. Of these, 19 had a positive SCOFF-F score (35.2%), 26 had a positive ROME-IV score (48.1%), and 14 patients (25.9%) had both a positive SCOFF-F score and a positive ROME-IV score (*p* = 0.006). Patients with positive SCOFF-F and ROME-IV scores had significantly higher HAD-anxiety and depression scores (*p* < 0.05). Conclusion: These results suggest a significant association between IBS, ED and endometriosis. The prevalence of IBS and ED in our population is higher than in the general population. Larger studies are needed to confirm these results, to better understand this triad, and to improve the diagnostic and multidisciplinary therapeutic management of these patients.

## 1. Introduction

Endometriosis affects 2.5 million women in France and 10% of all reproductive-aged women [1,2,3]. Endometriosis is characterized by the presence of endometrial tissue outside the uterine cavity. Different forms have been described: superficial or peritoneal, deep or sub-peritoneal, and ovarian endometrioma [1]. Deep endometriosis accounts for 6 to 10% of patients and 6 to 30% of these have digestive endometriosis. Endometriosis represents a real public health issue regarding infertility in 40% of women affected, negative impact on quality of life and psychological consequences (anxiety and depression) [1,2,3]. Two recent studies have shown a tight link between endometriosis and alteration of quality of life, anxiety and depressive disorders or stress [4,5]. In addition to gynecological symptoms (dysmenorrhea, dyspareunia), digestive disorders such as chronic abdominal pain and transit disorders are frequently reported in endometriosis [1,2,6,7,8,9]. This digestive symptomatology is also observed in patients with irritable bowel syndrome (IBS). IBS is a common functional bowel disorder that affects 15% of the western population, mainly women (70% are women), often diagnosed before the age of 50 [10,11]. Rome IV clinical criteria define IBS as the association of chronic abdominal pain and transit disorders [12,13]. A common symptomatology between IBS and endometriosis, often leads to misdiagnosis or delay in diagnosis of these two diseases, on average 6 to 7 years for endometriosis [2,3,6,7,8,9]. This delay is also related to the lack of awareness, and the absence of specific diagnostic tests either for IBS or endometriosis. In addition, chronic digestive disorders influence patients’ eating behavior, in an attempt to reduce these symptoms, often leading to food restriction [8,14,15,16]. Food restriction may lead to undernutrition that exacerbates digestive disorders, and also to eating disorders (ED) [14,15,16,17,18,19,20,21]. ED, including anorexia nervosa, bulimia, binge eating disorder, affect nearly 10% of the general population in France [22,23,24]. Moreover, both endometriosis, IBS and ED are often associated with anxiety and depression [10,13,15,21,25,26], because of chronic abdominal pain, digestive disorders, poor quality of life, and infertility in patients with endometriosis. Furthermore, several studies have identified common pathophysiological mechanisms between IBS and endometriosis [6,7,8,9], and also between IBS and ED [14,15,18,27], such as chronic low-grade inflammation, alterations of intestinal permeability, and gut microbiota dysbiosis [6,7,9,13,25,26,28,29,30,31,32,33,34,35,36,37,38]. However, to date, no study has assessed direct links between ED and endometriosis.

The aim of the ENDONUT pilot study was to assess associations between these three diseases by screening for IBS and ED in women with endometriosis.

## 2. Method

### 2.1. Population

Patients were recruited from the CIRENDO database (NCT02294825), the French North-West Inter-Regional Cohort of women with ENDOmetriosis approved in 2009 by the French authority CCTIRS (Advisory Committee on information processing in healthcare research, No 09.445). This prospective cohort has been registering women from 18 to 50 years with histologically documented endometriosis since June 2009 in 15 French centers (University Hospitals, public hospitals and private clinics).

Inclusion criteria were patients from the CIRENDO database with a recent documented diagnosis of endometriosis of less than 4 years (between 28 November 2018 and 24 June 2020, regardless of age, date of onset of symptoms, type of endometriosis (digestive or not), and with or without a digestive surgical procedure related to endometriosis. We included patients regardless of type of endometriosis and regardless of whether or not they had undergone endometriosis-related digestive surgery, and we assessed the influence of the presence of digestive endometriosis or a digestive surgical procedure related to endometriosis, in the response to the questionnaires. One hundred patients met these inclusion criteria, and all were from Rouen University Hospital

### 2.2. Data Collection

Patients were asked to participate in the ENDONUT pilot study by answering five validated questionnaires (Appendix A) sent to them by e-mail via a secure e-mail box at Rouen University Hospital: the SCOFF-F questionnaire for the screening of ED [23,24]; the EAT-26 questionnaire which is another questionnaire for the screening of ED; Rome IV criteria for the diagnosis of IBS [12,13]; the Francis score to assess the severity of IBS; the HAD scale for anxiety and depression. Anthropometric data and lifestyle habits were also collected (Appendix A). The ROME-IV and SCOFF-F scores that we used in our study, are validated scores used in many other studies [12,13,23,24]. The data collected were anonymized.

### 2.3. Statistical Analyses

The variables were analyzed in terms of numbers, means with standard deviations and percentages. Comparative analyses of qualitative variables were performed with Pearson’s Chi-square test or Fisher’s exact test, and comparative analyses of quantitative variables with Student’s T-test. We also performed non-parametric Mann–Whitney test and Kruskal–Wallis test followed by Dunn’s post-test. The main cross-tabulations were performed on the SCOFF-F score and ROME-IV score variables. A *p* value of <0.05 was considered statistically significant. Statistical analysis was performed using SPSS version 20.0 software (SPSS Inc., Chicago, IL, USA).

## 3. Results

The primary aim was to investigate an association between IBS, ED and endometriosis by screening IBS and ED in patients from the CIRENDO database. The primary composite endpoint was SCOFF-F and ROME-IV scores.

Secondary aims were to assess the influence of the presence of digestive endometriosis or a digestive surgical procedure related to endometriosis, in the response to the questionnaires; to assess the severity of IBS and the presence of associated anxiety-depressive disorders in different sub-groups. We chose as secondary endpoints the EAT-26 score, the Francis score, the HAD score and the data collected by anthropometric and lifestyle habits questionnaire.

Among the 100 patients meeting the inclusion criteria, 54 patients completed all questionnaires and were included in the final analysis (Figure 1).

The description of the patients, with the available data in the CIRENDO database, completed by the data collected from our questionnaires, is detailed in Table 1 and Table 2.

Among the 54 included patients, 19 had a positive SCOFF-F score (35.2%), 26 had a positive ROME-IV score (48.1%), and 14 patients (25.9%) had both a positive SCOFF-F score and a positive ROME-IV score (*p* = 0.006) (Figure 2).

We tested the ROME-IV score with several variables, and only the HAD-Anxiety and HAD-Anxiety and Depression variables were significantly different according to the response to the ROME-IV score (Table 3).

We performed the same tests with the SCOFF-F score, and only the HAD-Anxiety variable was significantly different according to the response to the SCOFF-F score (Table 4).

We completed the analysis with non-parametric Mann–Whitney test and Kruskal–Wallis test followed by Dunn’s post-test, and we found a significant difference in the response to the HAD questionnaire according to SCOFF-F status (Figure 3A), ROME-IV status (Figure 3B) and both SCOFF-F and ROME-IV (Figure 4). Patients with positive SCOFF-F and ROME-IV scores had significantly higher HAD-anxiety and depression scores (*p* < 0.05, Figure 3 and Figure 4).

We did not observe any difference between the type of endometriosis or whether or not patients had undergone endometriosis-related digestive surgery, in the response to the questionnaires (Table 5). Moreover, we did not observe any difference between the Francis score and other variables. However, the EAT-26 score was significantly different for the HAD-Anxiety and Depression variable.

## 4. Discussion

Our study suggests an association between IBS, ED and endometriosis. A quarter of patients had both a positive SCOFF-F score and a positive ROME-IV score (*p* = 0.006). Moreover, the prevalence of positive SCOFF-F and ROME-IV scores, in our population of patients with endometriosis, is higher than the prevalence of ED and IBS in the general population [11,22].

Interestingly, we found a difference in the number of positive SCOFF-F scores compared to EAT-26 scores. The SCOFF-F score, a rapid and simple screening tool used in primary care, screens typical and atypical forms of ED, with a high sensitivity and specificity; whereas the EAT-26 score, which is less easy to use in current practice, is very efficient for the detection of typical ED but less for atypical ED, leading to misdiagnosed patients.

We also found a significant difference in the response to the HAD questionnaire according to ROME-IV and SCOFF-F status, indicating the involvement of anxiety disorders in these diseases, in accordance with the literature [10,13,15,21,25,26]. In fact, ED, which are associated with a vulnerability to stress, as well as IBS and endometriosis, which are associated with chronic pain syndrome, have a significant psychological impact and affect quality of life, leading to the emergence of anxiety and depressive disorders, frequently associated with these diseases.

Among the proposed pathophysiological hypotheses, alterations of the gut–microbiota–brain axis seem to be one of the common mechanisms involved in the development of these diseases [6,7,9,25,26,28,29,30,32,33,37,38]. Several studies have shown significant changes in gut microbiota composition in IBS patients compared to healthy subjects [6]. Gut dysbiosis has been correlated both with the severity of IBS and with gut inflammation, while dysregulating the immune system and altering estrogen metabolism in patients with endometriosis [6,37,38]. Alterations of gut microbiota have also been reported in patients with ED [25,29,30,32,33]. Gut microbiota is a key actor of the gut barrier. Several studies reported alterations of intestinal permeability in IBS and endometriosis, leading to the translocation of bacterial endotoxins involved in the secretion of pro-inflammatory cytokines, and to a chronic low-grade inflammatory response [6]. In ED, few clinical studies have assessed the alterations of intestinal permeability, but increased intestinal permeability has been reported in a mice model of anorexia [32,34,35,36]. In addition, several studies have reported a link between alterations of the gut–microbiota–brain axis and mood regulation [25,26,39]. To date, no study has reported a direct link between ED and endometriosis, or between gut dysbiosis and endometriosis in humans [6,7,38]. In a recent review of the literature, the intestinal microbiota is described as an endocrine organ that can influence other organs and their signaling pathways, including the female reproductive organ [40]. Even though studies conducted on animal models suggest a link between the intestinal microbiota and endometriosis, no cause and effect relationship has been formally established between gut dysbiosis and endometriosis in humans. Nevertheless, understanding the mechanisms of this possible relationship could help in the development of preventive or therapeutic strategies.

The strengths of our study are (i) the inclusion of patients from the large CIRENDO cohort coordinated by an expert center; (ii) the representativity of the included patients regarding age, date of onset of symptoms, type of endometriosis, and whether or not they had undergone endometriosis-related digestive surgery; and (iii) the inclusion of patients from the CIRENDO cohort with a documented diagnosis of less than 4 years. The limits of our study are (i) the inclusion of patients only from Rouen University Hospital; (ii) among them nearly 50% had digestive endometriosis, and nearly 50% had undergone a digestive surgical procedure related to their endometriosis. We can explain this finding by the fact that all our patients were from Rouen University Hospital, an expert center in the field of endometriosis, which may have contributed to a selection bias. However, it does not seem to have affected our results, in fact, we did not observe any significant difference between the type of endometriosis or whether or not patients had undergone endometriosis-related digestive surgery, in the response to the questionnaires; however, as the presence of bowel endometriosis could lead to the over-estimation of IBS, it would be relevant to include patients with other types of endometriosis in future larger studies. (iii) A final limitation was the small population involved in this pilot study; only 54 patients completed all the questionnaires, among the 100 patients meeting inclusion criteria. We attribute this low response rate in part to the influence that the current health crisis may have had on patient participation in our study. However, it does not seem to have significantly altered our results, in fact, patients’ characteristics did not differ between «responders» and «non responders». Furthermore, we chose to include patients with a recent diagnosis of endometriosis, less than 4 years, and regardless of the date of onset of symptoms. Therefore, we do not have data regarding the time between diagnosis or onset of symptoms and completion of the questionnaires. But, to our knowledge, this is the first study to examine the overlap between these three diseases. Other large studies are needed to confirm these results, with a focus on gut microbiota composition and functions.

Finally, the ENDONUT pilot study suggests an association between irritable bowel syndrome, eating disorders, endometriosis and anxiety-depressive disorders. Further large studies should assess the common underlying mechanisms of this «four-leaf clover» (Figure 5). Screening for these diseases should be systematic, using validated tools. The aim is to avoid diagnostic and therapeutic errancy or delay, and propose multidisciplinary therapeutic management of these patients including, in addition to conventional medical management, nutritional and dietary management and psychological support. Furthermore, modulation of the gut–microbiota–brain axis through the patients’ diet, for example by enriching their diet with omega-3 fatty acids and dietary fiber, and limiting the consumption of ultra-processed foods and those rich in saturated fatty acids; or by proposing prebiotics or probiotics; or by performing a fecal microbiota transplantation, could constitute promising therapeutic perspectives in the management of these diseases.

## Figures and Tables

**Figure 1 jcm-11-05773-f001:**
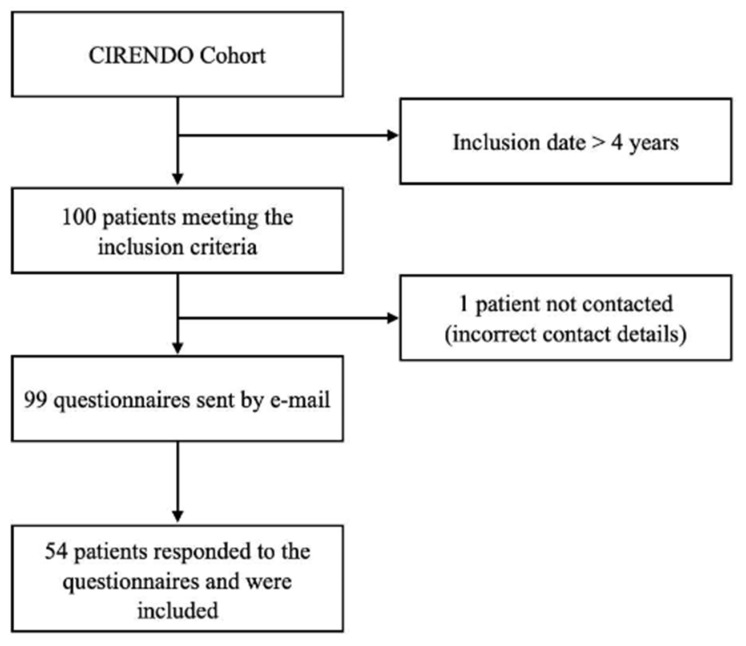
Flowchart for the selection of the study group.

**Figure 2 jcm-11-05773-f002:**
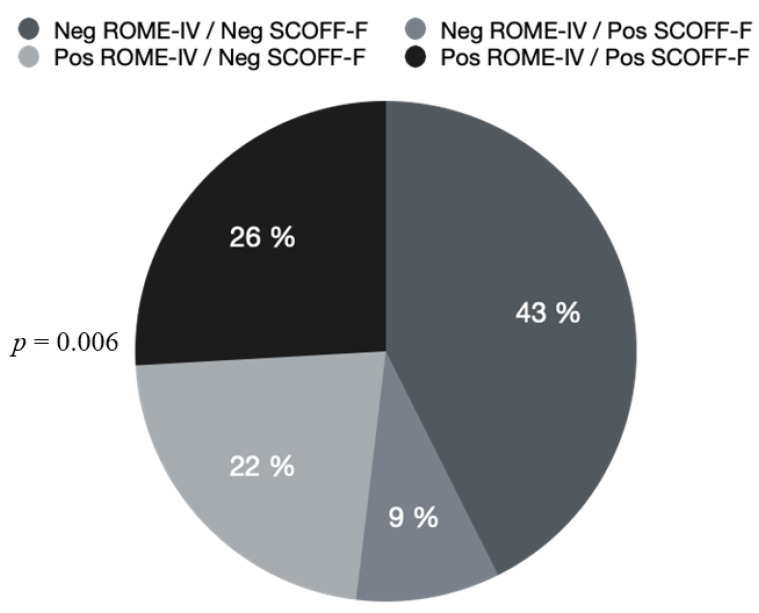
SCOFF-F and ROME-IV scores in the 54 responding patients. SCOFF-F: Sick, Control, One, Fat, Food-French version score; Neg ROME-IV: Negative ROME-IV; Neg SCOFF-F: Negative SCOFF-F; Pos ROME-IV: Positive ROME-IV; Pos SCOFF-F: Positive SCOFF-F.

**Figure 3 jcm-11-05773-f003:**
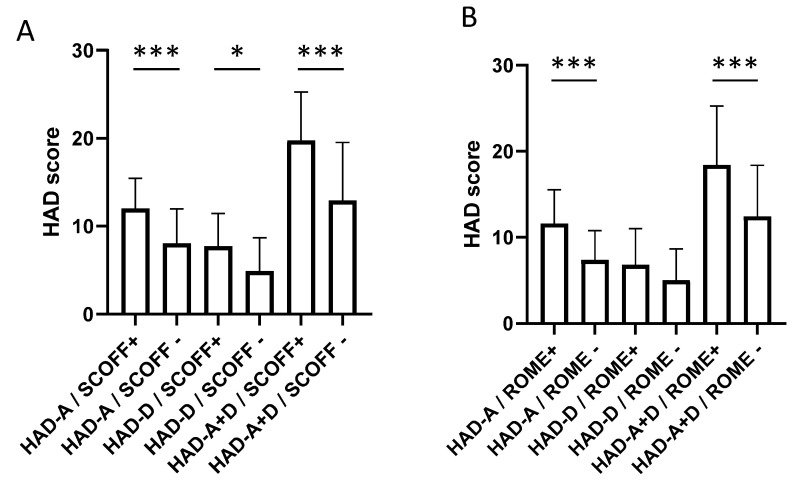
Response to the HAD questionnaire according to SCOFF-F status (**A**) and ROME IV status (**B**). *, *p* < 0.05; ***, *p* < 0.0001; HAD-A: HAD-Anxiety; HAD-D: HAD-Depression; HAD A + D: HAD-Anxiety and Depression; ROME+: ROME-IV positive; ROME−: ROME-IV negative; SCOFF+: SCOFF-F positive; SCOFF−: SCOFF-F negative.

**Figure 4 jcm-11-05773-f004:**
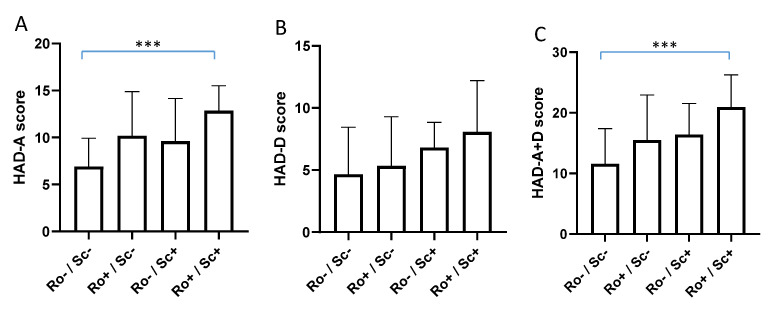
Response to the HAD questionnaire according to ROME-IV and SCOFF-F status combined. ***, *p* < 0.001 Dunn’s post-test; HAD-A: HAD-Anxiety (**A**); HAD-D: HAD-Depression (**B**); HAD A + D: HAD Anxiety and Depression (**C**); Ro+: ROME-IV positive; Ro−: ROME-IV negative; Sc+: SCOFF-F positive; Sc−: SCOFF-F negative.

**Figure 5 jcm-11-05773-f005:**
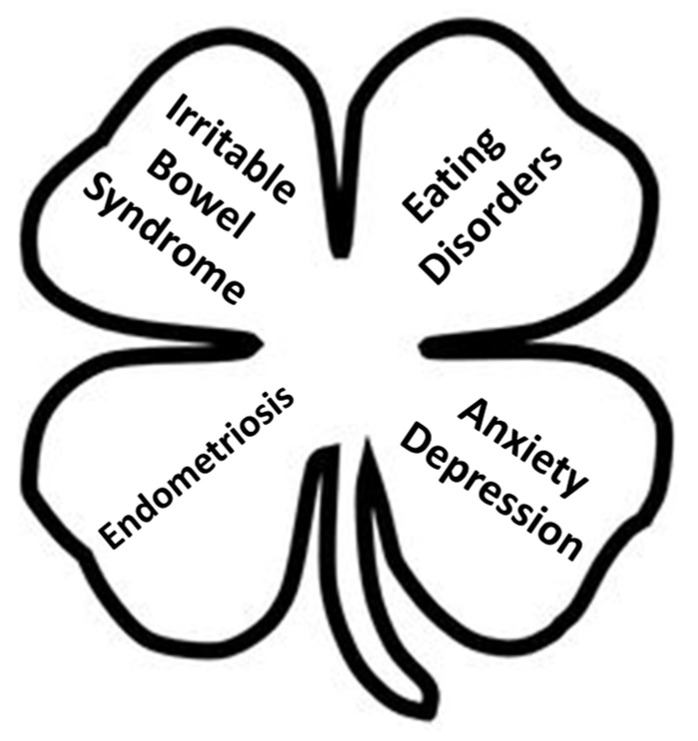
The four-leaf clover.

**Table 1 jcm-11-05773-t001:** Characteristics of the 100 patients meeting the inclusion criteria.

Variables	No (%)
Age (years)	(18–49) (34) 6.804 ^(1)^
Presence of digestive endometriosis	44 (48.4) ^(2)^
Endometriosis-related digestive surgery	44 (44)
Response to the questionnaires	54 (54)

^(1)^ (minimum–maximum) (mean) standard deviation. ^(2)^ (valid percentage) 9 missing data.

**Table 2 jcm-11-05773-t002:** Characteristics of the 54 responding patients included in the final analysis. BMI: Body Mass Index; IBS: Irritable Bowel Syndrome; ED: Eating Disorders; SCOFF-F: Sick, Control, One, Fat, Food-French version score; EAT-26: Eating Attitudes Test-26; HAD: Hospital Anxiety and Depression Scale.

Variables	No (%)
Age (years)	(18–49) (34.5) 7.427 ^(1)^
BMI (kg/m^2^)	(16.87–34.56) (24.85) 4.099 ^(1)^
Age at diagnosis of endometriosis (years)	(17–47) (31) 7.836 ^(1)^
Presence of digestive endometriosis	25 (50) ^(2)^
Endometriosis-related digestive surgery	26 (48.1)
History of IBS	10 (18.5)
History of ED	3 (5.6)
History of anxiety and depressive disorders	5 (9.3)
Delay in diagnosis of endometriosis > 6 years	38 (70.4)
Regular abdominal pain	35 (64.8)
Pain not related to the menstrual cycle	24/35 (68.6)
Regular use of analgesics	15 (27.8)
Regular use of anxiolytics	6 (11.1)
Ongoing hormone treatment for endometriosis	33 (61.1)
Positive SCOFF-F	19 (35.2)
Positive EAT-26	10 (18.5)
Positive ROME-IV	26 (48.1)
Positive HAD-Anxiety	33 (61.1)
Positive HAD-Depression	18 (33.3)
Positive HAD-Anxiety and Depression	12 (22.2)

^(1)^ (minimum–maximum) (mean) standard deviation. ^(2)^ (valid percentage) 4 missing data.

**Table 3 jcm-11-05773-t003:** ROME-IV score and other variables of interest in the 54 responding patients. EAT-26: Eating Attitudes Test-26; HAD: Hospital Anxiety and Depression Scale; *, *p* < 0.05.

	Negative ROME-IV No (%)	Positive ROME-IV No (%)	*p* Value (<0.05 *)
Positive EAT-26 No (%)	3 (5.6)	7 (13)	*p* = 0.169
Negative EAT-26 No (%)	25 (46.3)	19 (35.2)	
Presence of digestive endometriosis No (%) ^(1)^	15 (30) ^(1)^	10 (20) ^(1)^	*p* = 0.258
Absence of digestive endometriosis No (%) ^(1)^	11 (22) ^(1)^	14 (28) ^(1)^	
Endometriosis-related digestive surgery No (%)	14 (25.9)	12 (22.2)	*p* = 0.777
Absence of endometriosis-related digestive surgery No (%)	14 (25.9)	14 (25.9)	
Positive HAD-Anxiety No (%)	11 (20.4)	22 (40.7)	*p* = 0.001 *
Negative HAD-Anxiety No (%)	17 (31.5)	4 (7.4)	
Positive HAD-Depression No (%)	8 (14.8)	10 (18.5)	*p* = 0.441
Negative HAD-Depression No (%)	20 (37)	16 (29.6)	
Both positive HAD-Anxiety and Depression No (%)	2 (3.7)	10 (18.5)	*p* = 0.008 *
HAD-Anxiety and Depression both non-positive No (%)	26 (48.1)	16 (29.6)	

^(1)^ (valid percentage) 4 missing data.

**Table 4 jcm-11-05773-t004:** SCOFF-F score and other variables of interest in the 54 responding patients. SCOFF-F: Sick, Control, One, Fat, Food-French version score; HAD: Hospital Anxiety and Depression Scale; *, *p* < 0.05.

	Negative SCOFF-F No (%)	Positive SCOFF-F No (%)	*p* Value (<0.05 *)
Presence of digestive endometriosis No (%) ^(2)^	18 (36) ^(2)^	7 (14) ^(2)^	*p* = 0.239
Absence of digestive endometriosisNo (%) ^(2)^	14 (28) ^(2)^	11 (22) ^(2)^	
Endometriosis-related digestive surgeryNo (%)	19 (35.2)	7 (13)	*p* = 0.221
Absence of endometriosis-related digestive surgeryNo (%)	16 (29.6)	12 (22.2)	
Positive HAD-AnxietyNo (%)	17 (31.5)	16 (29.6)	*p* = 0.018 *
Negative HAD-AnxietyNo (%)	18 (33.3)	3 (5.5)	
Positive HAD-DepressionNo (%)	10 (18.5)	8 (14.8)	*p* = 0.314
Negative HAD-Depression No (%)	25 (46.3)	11 (20.4)	
Both positive HAD-Anxiety and DepressionNo (%)	5 (9.3)	7 (13)	*p* = 0.057
HAD-Anxiety and Depression both non-positiveNo (%)	30 (55.6)	12 (22.2)	
Francis score 0-199 No (%) ^(1)^	5 (19.2) ^(1)^	3 (11.5) ^(1)^	*p* = 0.508
Francis score 200-399 No (%) ^(1)^	6 (23.1) ^(1)^	10 (38.5) ^(1)^	
Francis score ≥ 400No (%) ^(1)^	1 (3.8) ^(1)^	1 (3.8) ^(1)^	

^(1)^ Only patients with positive ROME IV criteria are considered. ^(2)^ (valid percentage) 4 missing data.

**Table 5 jcm-11-05773-t005:** Type of endometriosis and endometriosis-related digestive surgery in the response or not to the questionnaires. *, *p* < 0.05.

	Absence of Response to the Questionnaires No (%)	Response to the Questionnaires No (%)	*p* Value (<0.05 *)
Presence of digestive endometriosis No (%) ^(1)^	22 (24.2) ^(1)^	25 (27.5) ^(1)^	*p* = 0.728
Absence of digestive endometriosis No (%) ^(1)^	19 (20.9) ^(1)^	25 (27.5) ^(1)^	
Endometriosis-related digestive surgeryNo (%)	18 (18)	26 (26)	*p* = 0.365
Absence of endometriosis-related digestive surgery No (%)	28 (28)	28 (28)	

^(1)^ (valid percentage) 9 missing data.

## Data Availability

The data presented in this study are available on request from the corresponding author.

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
