# Peer review of "Association between Endometriosis, Irritable Bowel Syndrome and Eating Disorders: ENDONUT Pilot Study"

_jcm, 2022, doi:10.3390/jcm11195773_

Round 1
Reviewer 1 Report (Previous Reviewer 1)
In the current paper, Aupetit and colleagues aimed at evaluating the correlation among irritable bowel syndrome (IBS), eating disorders (ED) and endometriosis, three pathologies sharing some common features and clinical manifestations.
The authors have significantly improved the quality of the manuscript as compared to the previous submission, addressing all the point raised before.
Author Response
Thank you very much for your comments and interest.
Reviewer 2 Report (Previous Reviewer 4)
Thank you for the opportunity to review the interesting study which demonstrates a link between IBD and endometriosis. However, there are some concerns which needs to be addressed.
Introduction section needs several additional sentences about impaired quality of life of this particular population. Please include the following, recently published references (PMID: 34718292 and PMID: 35819491)
Methodology and Results are clearly presented. However, in the Discussion section it would be great to focus more on a role of gut microbiota on endometriosis. Please provide essential information on what preventive or therapeutic dietary changes can be considered.
Author Response
Please find attached the revised manuscript checked by a colleague fluent in English writing as per the editor’s comments.
Reviewer 2
Thank you very much for your new comments and suggestions to improve our manuscript.
As you suggest, we have developed in the introduction the point about the quality of life of this particular population and included the two references mentioned (line 57) :
- PMID 34718292 : Skegro B, Bjedov S, Mikus M, Mustac F, Lesin J, Matijevic V, et al. Endometriosis, Pain and Mental Health. Psychiatr Danub 2021;33:632-636
- PMID 35819491 : Mikus M, Matak L, Vujic G, Skegro B, Skegro I, Augustin G, et al. The short form endometriosis health profile questionnaire (EHP-5): psychometric validity assessment of a Croatian version. Arch Gynecol Obstet 2022 Jul 11
In addition, as suggested, in the Discussion section, we have also developed the role of the gut microbiota in endometriosis with regards to recent studies (line 261). Finally, we completed the information regarding dietary management (line 314).
Round 2
Reviewer 2 Report (Previous Reviewer 4)
N/A
This manuscript is a resubmission of an earlier submission. The following is a list of the peer review reports and author responses from that submission.
Round 1
Reviewer 1 Report
In the current paper, Aupetit and colleagues aimed at evaluating the correlation among irritable bowel syndrome (IBS), eating disorders (ED) and endometriosis, three pathologies sharing some common features and clinical manifestations. The study, despite being interesting in its purposes, suffers of lack of clarity in some points and a very limited amount of data.
Introduction: introduction is overall well written and complete enough. However, it results a little bit unclear in some sentences: for instance, in lines 54-55, authors should motivate and distinguish information about IBS and endometriosis late diagnosis. In lines 61-64 authors should also be more clear and detailed in their statements.
Methods:
- Lines 84-86: the description of the questionnaires that were used is quite confused. For instance, Francis score is about IBS severity, but from the text it seems related to anxiety and depression. Also EAT-26 is not clear if it is about ED, anxiety and depression or both. Authors should reformulate the sentences in favor of an easier understanding for the readers.
- The paragraph about statistical analysis contains information not appropriate to methods section. All the lines 89-97 should be moved to Results or Discussion section.
Results:
- In the Results section figures and table should be described more accurately. This would allow the authors to create longer Result comments interspersed with the associated table or figure. Everything seems quite approximate in the current form.
- Tables are quite confusing, in particular as regard table legends.
- May be possible to include in your study some healthy controls? In this way your hypothesis would appear stronger and less dependent on the bias of your cohort selection.
- Line 112: authors claimed that it was not possible to find a correlation regarding the type of endometriosis, but what about different stages of the disease?
- The assertion that ED and endometriosis may be associated is not supported by data but just an overstatement, since no correlation seemed to be found in the manuscript. This should be discussed more clearly.
- Have you tried to analyze the data with regard to the different time elapsed after diagnosis and filling out of the questionnaire? The first months after diagnosis may be determinant in modifying food habits and experiencing anxiety and depression. Moreover, also pregnancy or infertility may be a determinant factor in anxiety and depression. In any case, both endometriosis and the other considered pathologies shares important and not easily evaluable psychological implications, such as increased stress, that may induce a symptom overlap. This aspect should be highlighted in the Discussion.
Discussion:
- Maybe authors may report other studies in which these questionnaires were used in order to strengthen their validation and usefulness, also because not all the readers are familiar with those kinds of questionnaires.
- Discussion should be reconsidered on the basis of the previous considerations about the reported results since in the current form is not so convincing.
Author Response
Please find attached our reply

Reviewer 2 Report
The authors conducted a study to assess the association between endometriosis on the one hand and irritable bowel syndrome, eating disorders, and anxiety on the other hand. This study has serious methodological flaws. Firstly, the authors included patients with bowel endometriosis and patients that had bowel surgery for endometriosis. It is well-known that digestive endometriosis causes symptoms similar to IBS, which may lead to over-diagnosing IBS in this sample. We also do not know how the bowel function after surgery for endometriosis. Secondly, the included sample is very small and unrepresentative. Moreover, almost half of the sample had endometriosis and bowel surgery, which may lead to a biased estimation of the IBS and eating disorders prevalence. Lastly, the Chi-square test is not a good option to measure the association between variables. This could have been better done with regression models or correlation tests.
Comments:
1) In the results section of the abstract (line 33): it is not clear what does the written p-value represent? What did you compare? and which groups did you compare?
2) In the introduction section of the main text (line 54): I agree that shared symptoms between endometriosis and other diseases contribute to the diagnostic delay of endometriosis but the phrase in its current form attributes the diagnostic delay entirely to this factor. Lack of awareness and non-invasive diagnostics of endometriosis are important factors influencing the diagnosis. These should not be neglected. Please revise this part accordingly.
3) In the population paragraph (lines 78-79): please explain why you included patients with bowel endometriosis and patients that had an operation for gastrointestinal endometriosis.
4) The time of inclusion is not written correctly in figure 1.
Author Response
please find attached our reply

Reviewer 3 Report
We congratulate the authors for their article. We always felt the association between bIBS and endometriosis exist.
Now the authors proved it scientifically.
The manuscript is mature enough to be accepted and published.
No suggestions / edits are needed.
Author Response
please find attached our reply

Reviewer 4 Report
This is an interesting and very novel pilot study regarding association between endometriosis, IBS and eating disorders. Here are some suggestions which I made in order to improve overall manuscript quality:
- it will be great to provide an accurate information for annual occurrence of IBS and eating disorders in France, if possible (as it was stated for endometriosis)
- it will be useful to put a brief description of used questionnaires in the Methods section (with appropriate references). Are the used questionnaires validated on French population?
- Please provide essential information on what preventive or therapeutic dietary changes can be considered (in Discussion section)
Author Response
please find attached our reply
Reviewer 5 Report
This is a very primitive study based on only 54 patient data with the completed questionnaire. The conclusions made out of this study are huge and can make a big impact in this field of study. A study with a higher number of patients divided into groups based on type and stage of endometriosis is required.
Author Response
We thank you for your comments and agree that a largest study is needed.
Round 2
Reviewer 1 Report
The manuscript has been quite improved after first revision step. I suggest to carefully check English and the general quality of the writing in the final version.
Author Response
We thank the reviewer for the comments and suggestions to improve our manuscript.

Reviewer 2 Report
Dear Authors:
Thank you for your efforts in revising your manuscript. Please find my comments below.
1) In response to your comment: "We assessed the influence of the presence of digestive endometriosis or a digestive surgical procedure related to endometriosis, in the response to the questionnaires. There was no significant difference between groups «digestive endometriosis, or not » and «digestive surgical procedure related to endometriosis, or not »".
The small sample size is not indicative, and therefore, the analysis lacks statistical power. Besides, the presence of bowel endometriosis could lead to the over-estimation of IBS. It would have been better to include patients with other types of endometriosis except for GI endometriosis.
2) In response to your comment: "The p-value presented in the Results section of the abstract represents the significant association between IBS and ED in 14 endometriosis patients with both positive response to the ROME-IV and SCOFF-F scores".
The Chi-square test does not measure association, its main use comparing ratios between two groups to see whether the observed differences are significant or merely attributed to chance. In other words, it is a test that compares percentages between two groups, not the association. In the abstract and your response, it is not clear which two groups you compared and obtained statistical significance. Besides, a significant Chi-square test does not indicate correlation. Please pay special attention to this point.
3) In response to your comment: "We chose to include patients regardless of type of endometriosis (digestive or not) and regardless of whether or not they had undergone digestive surgery related to endometriosis, because we wanted to have a mixed population of patients with endometriosis".
I agree that the bigger the sample is, the more representative it becomes. But in such situations, special attention should be paid to avoid the confounding effect of including patients with different types of pathologies and surgical procedures. In this research, the inclusion of these two groups would only confuse the results. It is very likely that digestive endometriosis and bowel surgery alter the bowel function and may lead to over-estimation of IBS. For this reason, the relationship between endometriosis and IBS should have been assessed in a population that did not have bowel endometriosis and bowel surgery to have a more precise estimate of the correlation.
Author Response
Reviewer 2 :
Thank you very much for your new comments and suggestions to improve our manuscript.
1) In response to your comment: "We assessed the influence of the presence of digestive endometriosis or a digestive surgical procedure related to endometriosis, in the response to the questionnaires. There was no significant difference between groups «digestive endometriosis, or not » and «digestive surgical procedure related to endometriosis, or not »".
The small sample size is not indicative, and therefore, the analysis lacks statistical power. Besides, the presence of bowel endometriosis could lead to the over-estimation of IBS. It would have been better to include patients with other types of endometriosis except for GI endometriosis.
Indeed, we are aware of the small size of our sample and this is why we insist on the fact that this is a pilot study, very primitive. Moreover, to our knowledge, this is the first study to investigate the overlap between these 3 pathologies. We hope that it will lead to perspectives for patient management and for future large studies. In fact, studies with a larger number of patients and patients with different endometriosis profiles are necessary. We added the reviewer’s comment in the discussion section (limits: line 210-212).
2) In response to your comment: "The p-value presented in the Results section of the abstract represents the significant association between IBS and ED in 14 endometriosis patients with both positive response to the ROME-IV and SCOFF-F scores".
The Chi-square test does not measure association, its main use comparing ratios between two groups to see whether the observed differences are significant or merely attributed to chance. In other words, it is a test that compares percentages between two groups, not the association. In the abstract and your response, it is not clear which two groups you compared and obtained statistical significance. Besides, a significant Chi-square test does not indicate correlation. Please pay special attention to this point.
All the variables (SCOFF-F, Rome-IV, EAT-26, HAD ...) were classified as qualitative variables (yes/no). In fact, we performed a chi-square test to compare percentages between 2 qualitative variables. As indicated by the reviewer, it is not an association but the representation of a significant difference between two groups ; here, among patients with endometriosis, we tested two by two the Rome-IV group (yes/no: meeting IBS criteria or not) and the SCOFF-F group (yes/no; meeting ED criteria or not) and we highlighted a significant difference between the groups ; those who met the IBS criteria were also statistically those who met the ED criteria and vice versa.
As suggested, we reformulate for clarity both in the abstract, the results, the discussion and Figures. Lines 32-34, 134-136, 141-143, 148-149, 154-158, 167.
3) In response to your comment: "We chose to include patients regardless of type of endometriosis (digestive or not) and regardless of whether or not they had undergone digestive surgery related to endometriosis, because we wanted to have a mixed population of patients with endometriosis".
I agree that the bigger the sample is, the more representative it becomes. But in such situations, special attention should be paid to avoid the confounding effect of including patients with different types of pathologies and surgical procedures. In this research, the inclusion of these two groups would only confuse the results. It is very likely that digestive endometriosis and bowel surgery alter the bowel function and may lead to over-estimation of IBS. For this reason, the relationship between endometriosis and IBS should have been assessed in a population that did not have bowel endometriosis and bowel surgery to have a more precise estimate of the correlation.
Indeed, in our study, the large number of patients with digestive endometriosis and/or having undergone a digestive procedure related to their endometriosis is not negligible, although we did not find any significant difference between the groups in the response to the SCOFF-F and ROME-IV scores otherwise. Nevertheless, we insist on this limit line 215-217 as suggested by the reviewer.
